# Chronic Stress, Exercise and Cardiovascular Disease: Placing the Benefits and Risks of Physical Activity into Perspective

**DOI:** 10.3390/ijerph18189922

**Published:** 2021-09-21

**Authors:** Barry A. Franklin, Akash Rusia, Cindy Haskin-Popp, Adam Tawney

**Affiliations:** 1Preventive Cardiology and Cardiac Rehabilitation, Beaumont Health, Royal Oak, MI 48073, USA; cindy.haskin-popp@beaumont.org; 2Internal Medicine and Biomedical Engineering, Oakland University William Beaumont School of Medicine, Rochester, MI 48309, USA; 3Cardiovascular Medicine, Department of Cardiology, Beaumont Health, Royal Oak, MI 48073, USA; akash.rusia@beaumont.org (A.R.); adam.tawney@beaumont.org (A.T.)

**Keywords:** physical activity, chronic stress, atherosclerotic cardiovascular disease, acute myocardial infarction, sudden cardiac death, hypertrophic cardiomyopathy, coronary artery calcium, atrial fibrillation

## Abstract

Chronic stress, which has been exacerbated worldwide by the lingering COVID pandemic, has been strongly linked to cardiovascular disease (CVD). In addition, autonomic dysregulation via sustained sympathetic activity has been shown to increase the risk of arrhythmias, platelet aggregation, acute coronary syndromes and heart failure. Fortunately, effective coping strategies have been shown to attenuate the magnitude of hyperarousal associated with the stress response, including moderate-to-vigorous lifestyle activity and/or structured exercise. A good-to-excellent level of cardiorespiratory fitness also appears to be highly cardioprotective. These beneficial effects have been substantiated by numerous studies that have evaluated the levels of stress reactivity and stress recovery in physically active individuals versus matched sedentary controls, as well as before and after exercise interventions. On the other hand, unaccustomed strenuous exercise in habitually sedentary persons with underlying CVD is associated with a disproportionate incidence of acute cardiac events. Moreover, extreme exercise regimens appear to increase coronary calcification and the likelihood of developing atrial fibrillation. This review summarizes these relations and more, with specific reference to placing the benefits and risks of physical activity into perspective.

## 1. Introduction

Psychosocial stress encompasses various stressors that can be categorized into three general groups: major life stress (e.g., occupational, relationship, caregiving, death of a loved one), socioeconomic factors (e.g., low income, discrimination, crime), and chronic psychiatric conditions (e.g., depression, anxiety). Numerous studies over the last several decades have demonstrated that psychosocial stress adversely affects many organ systems in the body [1,2,3,4,5]; however, particularly concerning is its effect on cardiovascular disease (CVD) and mortality. The landmark INTERHEART study demonstrated that psychosocial factors carry an odds ratio for myocardial infarction similar to that of “traditional” risk factors [6]. Their effect on other CVDs appears similar.

### 1.1. Major Life Stress

High levels of work or occupational stress have been convincingly linked to a greater risk for hypertension and cardiovascular mortality [7,8]. One model that correlates with coronary disease as well as psychological symptoms describes the variables at play which contribute to “job strain,” specifically the levels of demand and control [9]. Jobs with high demands and low control are associated with elevated blood pressure at home, work, and during sleep. Chronic hypertension is known to cause pathologic remodeling and is substantiated by a connection between job stress and left ventricular mass [9,10], perhaps providing a barometer for the chronicity and magnitude of hypertension elicited. Marital or relationship stress is likely of greater significance in relation to cardiovascular risk given its inherent nature in peoples’ lives. Several studies have reported that marital strife is associated with the development and/or progression of atherosclerotic CVD as well as cardiovascular mortality with a comparable risk to that of cigarette smoking or physical inactivity [11,12,13]. Others have linked marital stress in women directly with coronary atherosclerosis determined by angiographic studies rather than merely cardiovascular risk factors [14]. Similarly, caregiving stress appears to have a heightened effect on the risk of CVD in women. The Nurses’ Health Study showed that caregiving (>8 h/week) for a disabled or ill spouse was associated with an increased relative risk (1.82) of coronary heart disease [15]. Interestingly, the findings did not show the same effect when caregiving for a disabled or ill parent.

### 1.2. Socioeconomic Factors

Although socioeconomic status (SES), including factors such as financial hardship, housing conditions, employment, and education, is a reported mediator of cardiovascular risk [16,17,18,19], it is difficult to characterize and quantify, because it is strongly associated with other potential confounders such as chronic life stress [20] and depression [21]. Education is considered the strongest risk predictor of the many dimensions of SES [22] and the easiest to measure. The Northern Manhattan Study evaluated the relation of SES and race with left ventricular mass, noting that low SES (defined by education level) is an independent predictor of increased left ventricular mass among hypertensive and normotensive blacks, but not in whites or Hispanics [23]. More controversial is the potential link between racial discrimination and CVD with studies showing mixed results [24,25,26,27]. Nevertheless, favorably modifying certain socioeconomic factors can decrease cardiac risk with one study showing that alleviating financial stress was associated with reduced ambulatory blood pressures [28].

### 1.3. Psychiatric Conditions

Superimposed depressive disorders, ranging from mild or subclinical depressive symptoms to classic major depression, are associated with a heightened risk of coronary events in patients with and without coronary artery disease [29]. Moreover, in post-myocardial infarction patients, there is a relation between the severity of depression and survival [30]. Accordingly, major depression highly correlates with subsequent cardiac events [31]. Similar to depression, anxiety is also considered a strong predictor of acute cardiac events in patients with stable coronary disease [32]. More commonly associated with chronic psychosocial stress, post-traumatic stress disorder increases the risk of CVD independent of familial history and documented psychiatric comorbidities [33,34,35,36]. More severe psychiatric diseases such as schizophrenia carry a nearly two-fold higher risk of cardiovascular mortality than in the general population [37]. The reasons for the increased mortality may also be partially attributed to potential confounders, including the high rate of cigarette smoking and obesity in patients with schizophrenia [38,39,40]. Additionally, the antipsychotic drugs commonly prescribed for schizophrenia and other psychotic diseases are associated with significant weight gain and the development of diabetes and dyslipidemia [41,42].

Although chronic psychosocial stress appears to represent a strong risk factor for CVD, the underlying mechanisms require clarification. Specifically, while some stress disorders likely promote unhealthy lifestyle habits, escalating data suggest that chronic stress can independently lead to increased rates of CVD and mortality through varied pathophysiologic mechanisms. 

### 1.4. Pathophysiology

Stressors, whether intrinsic or extrinsic, can adversely impact homeostasis. The key pathways responsible are the hypothalamic–pituitary–adrenal (HPA) axis and the autonomic nervous system. These systems, along with other centers in the central nervous system, constitute complex psycho–neural–immuno–endocrine–adrenal pathways that interact with tissue and organ systems to respond to stressors. Chronic stress can cause dysregulation of these systems, disrupting homeostasis and leading to a spectrum of detrimental clinical manifestations involving the cardiovascular system [43,44]. 

Stress mediated activation of the hypothalamic paraventricular nucleus causes secretion of corticotropin-releasing factor. This causes secretion of adrenocorticotropic hormone from the anterior pituitary gland. In response, cortisol is released into the bloodstream from the adrenal cortex. Negative feedback of cortisol on the hypothalamus and anterior pituitary maintains control of the axis. Chronic stress leads to an imbalance in this regulation through a cumulative increase in basal cortisol levels via the HPA axis [45]. Clinical studies have reported that this imbalance can lead to atherosclerotic plaque formation, hypertension, insulin resistance, central adiposity, or combinations thereof [46]. 

The autonomic nervous system, comprised of a network of neurons in the brainstem, spinal cord and heart primarily modulates heart rate, myocardial contractility and vascular tone via direct neural connections and circulating catecholamines. It has two balancing systems, the sympathetic and parasympathetic nervous system. The sympathetic nervous system is particularly utilized during physical exertion and stress (increased heart rate, myocardial contractility, vascular tone). The parasympathetic system is active under conditions of rest and counteracts the sympathetic response [47]. Chronic stress causes autonomic dysregulation via sustained sympathetic activity and/or reduced parasympathetic activity [48]. This shift has been shown to increase the risk of arrhythmias, platelet aggregation, acute coronary syndromes and heart failure [44,49].

## 2. Impact of Physical Activity/Exercise on the Stress Response

The stress response is a primitive mechanism triggered by a threat, real or perceived, to one’s well-being and survival. It is characterized by a cascade of protective physiological and psychological responses that prepare the individual to either combat the threat or flee from it. The stress response serves the individual well when the threat is temporary. However, health problems often arise when the threat persists, or if other threats arise concurrently or in rapid succession, creating a state of chronic stress. 

Chronic stress is a risk factor for various physical and mental health conditions, including CVD, certain cancers, hypertension, diabetes, sleep disturbances, digestive disorders, upper respiratory disease, immune suppression, mood swings, anxiety, and depression [50]. Moreover, stress has been linked to an increased incidence of drug overdose and suicide across all ages, which has contributed to a recent decrease in the average life expectancy of Americans [51,52]. 

### 2.1. Prevalence of Stress Today

Unfortunately, chronic stress has become increasing prevalent throughout the world. According to the *Gallup Global Emotions 2020 Report* (conducted pre-COVID-19 pandemic), over one-third of people worldwide reported experiencing escalating levels of stress [53,54]. More recently, stress associated with the COVID-19 pandemic has led to increases in the incidence of psychological distress. The findings presented in the *Stress in America 2020 Report* compared to the *2019 Annual Stress in America Survey* indicated that the average level of reported stress for U.S. adults was significantly higher in 2020 than it was the previous year (5.4 vs. 4.9 on a scale of 1 [very low] to 10 [very high], respectively) [55,56]. In addition, the average level of stress related to the COVID-19 pandemic was 5.9 on this scale. Furthermore, there was a 21% increase in the number of anti-depressant, anti-anxiety, and anti-insomnia medication prescriptions filled during the initial weeks of the realization of the pandemic, 16 February–15 March 2020, with 78% of those prescriptions being new, according to findings presented in *America’s State of Mind Report* [57]. The authors of the report suggested a correlation with the COVID-19 pandemic. 

With stress levels on the rise in response to COVID-19, coping strategies have increasingly become a public health focus. Effective coping strategies can offset the state of hyperarousal associated with the stress response, whereas suboptimal coping skills can lead to physical and mental health problems over time, even if they do provide the individual with some immediate, temporary relief. Key modulators have been described by the stress bucket analogy (Figure 1) which originated from the pioneering work of Brabban and Turkington [58].

### 2.2. Stress Bucket Analogy

In the stress bucket analogy, the bucket represents an individual’s level of stress tolerance. The size of the bucket largely depends upon a person’s physiology, personality, and life experiences. Everyone has a different size bucket, which remains unchanged. The water level in the bucket represents the sum of all stresses in an individual’s life at that time. People who have larger buckets can hold more water (i.e., have a greater tolerance for stress) than those with smaller buckets. 

If the bucket overflows, health problems often develop, including CVD. To keep the water in the bucket from overflowing, individuals use the coping strategies readily available to them. Coping strategies act similar to taps in the bucket to drain the water. Effective taps (e.g., healthy lifestyle choices, such as adequate sleep and regular physical activity [PA]) decrease the amount of water in the bucket and relieve stress. Faulty taps (e.g., alcohol and drug abuse) provide temporary relief, but ultimately result in more water flowing into the individual’s bucket. Physical and mental health problems develop when the water in the bucket repeatedly overflows.

### 2.3. Coping Methods

Historically, stress management techniques have included meditation, deep breathing exercise, mental imagery, progressive muscle relaxation, and in some cases psychotherapy and/or pharmacologic treatment. In recent years, the role that moderate-to-vigorous PA plays in stress management has gained increasing attention.

It is well documented that regular PA has a positive impact on general health and well-being. There is also growing evidence to suggest that both single and cumulative bouts of PA can elicit significant stress reducing effects (Figure 2). Furthermore, some studies have suggested that a high level of cardiorespiratory fitness (CRF) may favorably modulate the response to chronic stress [59,60,61,62,63].

### 2.4. Physical Activity: Exercise Training and Stress Reduction

One randomized controlled study investigated the physiological response to psychosocial stress in healthy men (*n* = 96) [59]. Subjects were divided into three study groups: (1) an intervention group that underwent 12-weeks of endurance training; (2) an attention control group that underwent a 12-week relaxation program; and (3) a waitlist control group that received no intervention. 

At baseline or study entry, the men did not differ in body mass index, chronic stress, trait anxiety, activities of daily living, habitual level of PA or level of fitness. The subjects’ endocrine stress response (HPA axis) was determined by measuring salivary free cortisol and the autonomic stress response was assessed by measuring heart rate and heart rate variability. The responses were defined by level of *stress reactivity* (degree of response to stressor exposure) and *stress recovery* (how long the variable remained elevated after *stressor cessation*).

Results showed that the subjects who underwent 12-weeks of endurance exercise training had significantly reduced stress reactivity to a psychosocial stressor as measured by cortisol, heart rate, and heart rate variability responses. The subjects who underwent 12-weeks of relaxation technique training demonstrated reductions in cortisol stress reactivity only and those in the wait list group showed no changes in stress reactivity as measured by the standardized protocol. Furthermore, only those subjects who underwent endurance training had significant reductions in heart rate reactivity and improvements in heart rate recovery (i.e., how long the heart rate remained elevated after cessation of the psychosocial stressor).

It should be noted, however, that reductions in cortisol stress reactivity were not significantly different between the endurance training and relaxation training groups. However, only the endurance training group experienced significant reductions in cortisol stress reactivity compared to the wait list control group, whereas the difference in cortisol stress reactivity between the relaxation training and wait list control groups only approached significance. 

Although exercise levels increased significantly for both the endurance exercise training and relaxation technique training groups, only those subjects in the exercise group demonstrated significant improvements in CRF. The researchers concluded that there is a causal relationship between PA and the physiological response to stress, with endurance exercise training resulting in reductions in reactivity to a standardized psychosocial stressor as measured by cortisol, heart rate and heart rate variability responses.

It has been suggested that regular PA has a protective effect against stress. Rimmele et al. [60] investigated the endocrine and autonomic responses to a psychosocial stressor among 22 elite sportsmen as compared with 22 healthy untrained men. The subjects’ salivary free cortisol levels, heart rate and psychological responses as defined by mood, calmness and anxiety were measured before and after the stressor (Trier Social Stress Test). The elite sportsmen demonstrated lower cortisol and heart rate responses (main effect of group (*F*(1,39) = 5.47, *p* < 0.05 and (*F*(1,37) = 7.27, *p* < 0.05, respectively), better mood (group by time interaction effect *F*(1,39) = 5.80, *p* < 0.05), greater calmness (main effect of group (*F*(1,39) = 16.45, *p* < 0.1), and lower state anxiety (*F*(1,38) = 15.55, *p* <0.001) compared with the untrained men. There were no significant differences in cortisol levels, heart rate, or mood at baseline between the two groups. The elite sportsmen demonstrated significantly higher levels of calmness and trended towards lower levels of state anxiety at baseline. It was concluded that regular PA may offer a protective effect against stress and its related disorders (anxiety) [60].

Another study assessed the effects of varied PA interventions on the psychological well-being of 147 adolescents [61]. Subjects were assigned to one of four groups: (1) high-intensity aerobic training; (2) moderate-intensity aerobic training; (3) flexibility training; and (4) a control group. All subjects completed baseline and follow-up assessments of stress, well-being, and exercise tolerance. These assessments, along with measurements of the subjects’ fitness levels and heart rate and blood pressure responses, were obtained at the beginning and end of the 10-week investigation.

Greater levels of PA were associated with lower levels of stress and depression based on analysis of the subjects’ self-reports. Adolescents assigned to the high-intensity aerobic training group reported experiencing significantly less stress compared with subjects in the other three groups. In addition, only those subjects in the high-intensity aerobic training group reported reduced levels of stress and anxiety, depression, and hostility at the end of the investigation, whereas subjects in the other groups trended in the opposite direction. The researchers concluded that high-intensity exercise is an effective intervention to enhance adolescent well-being.

Other investigators have reported that mild-to-moderate intensity exercise can also provide protective effects against stress. Rogers et al., [62] found that exercise at 40–50% maximal oxygen uptake resulted in reductions in resting blood pressure and an attenuated blood pressure response to stress in borderline hypertensive individuals.

### 2.5. Physical Activity: Acute Exercise and Stress Reduction

Evidence also suggests that even single bouts of PA can be used to transiently mitigate the effects of stressful situations. Wunsch et al., [63] evaluated the effects of both habitual and acute exercise on the responses to psychosocial stress as measured by endocrine and autonomic stress response system biomarkers, specifically free salivary cortisol and alpha amylase. Subjects included 84 males, aged 18–30 years, half of whom were habitual exercisers whereas the remaining participants were generally inactive at baseline. The men were randomly assigned to one of two groups: an acute exercise intervention group who engaged in 30 min of moderate-to-high-intensity cycle ergometer exercise and a control group who participated in 30 min of light stretching exercise. Both groups were exposed to a psychosocial stressor (Trier Social Stress Test for Groups) before and after the intervention. Samples of each subject’s saliva were obtained to assess free salivary cortisol and alpha amylase levels. It was concluded that both habitual and acute forms of exercise can have a favorable impact on modifying the response to a standardized psychosocial stressor

### 2.6. Exercise Prescription to Reduce Stress

Research-based definitive guidelines for prescribing exercise to reduce chronic stress are currently unavailable. Various factors may play a role in determining the impact that regular PA has on the stress response, such as the training intensity, level of CRF, co-morbidities/medical history, magnitude of life stress, and the individual’s exercise preferences. Nevertheless, the general consensus is that the total volume of bodily movement or simply being physically active throughout the day plays a greater role in protection than any one exercise parameter (i.e., frequency, intensity, duration, and mode) [52]. Therefore, following the American College of Sports Medicine’s guidelines of engaging in 150 min/week of moderate-intensity PA, or 75 min/week of vigorous PA, or combinations thereof, should provide sufficient levels of exercise to promote overall physical and mental health, including stress reducing effects.

## 3. Exercise/Physical Activity: A Double-Edged Sword

Exercise training, as a subcategory of PA, is defined as any structured intervention with the objective of increasing or maintaining CRF or health, enhancing athletic performance, or both. Peak oxygen consumption or CRF can be directly measured during cardiopulmary exercise testing or estimated from the attained treadmill or cycle ergometer workload, adjusted for duration. Although considerable epidemiologic evidence suggests that regular moderate-to-vigorous PA may help to reduce chronic stress and protect against the development of atherosclerotic CVD, exertion-related acute cardiac events have been reported in the medical literature [64] as well as the lay press [65], suggesting that vigorous PA (≥6 metabolic equivalents [METs]; 1 MET = 3.5 mL/kg/min) may trigger cardiac arrest or acute myocardial infarction (AMI) in persons with known or occult CVD [66]. Several triggering mechanisms for plaque rupture and acute coronary thrombosis (Table 1) [67] and threatening ventricular arrhythmias (Figure 3) have been suggested [68].

Structural cardiovascular abnormalities, especially hypertrophic cardiomyopathy (HCM), are the major causes of exertion-related sudden cardiac death (SCD) in younger athletes [69]. In contrast, atherosclerotic CVD is the most common autopsy finding in middle-aged and older adults [70]. In a landmark study from the Cleveland Clinic, investigators estimated that ~85% of individuals in the U.S. ≥ 50 years of age have subclinical evidence of coronary artery disease [71]. Thus, it is the combination of vigorous PA and atherosclerotic or structural heart disease, rather than the exercise per se, that seems to present the trigger for cardiac events associated with strenuous physical exertion.

The relative risk (RR) for acute cardiac events during or immediately after mild-to-moderate intensity exercise is similar to that expected by chance alone. In persons with known or occult CVD, high-volume, high-intensity training regimens or competition seem to be associated with a heightened incidence of acute cardiovascular events [72]. The absolute risks of exercise-related cardiovascular events in apparently healthy adults are 1 per 1,124,200 and 1 per 887,526 person-hours for nonfatal and fatal events, respectively [73]. Thus, strenuous bouts of PA, particularly when unaccustomed, may transiently increase the risk of cardiovascular complications; however, the absolute risk associated with each exercise session remains extremely low.

Although AMI and SCD can be triggered by vigorous PA, the risk decreases with increasing frequencies (days/week) of vigorous PA. The RR seems to be highest for inactive individuals with known or occult CVD who were performing unaccustomed vigorous PA. For example, in the Determinants of Myocardial Infarction Onset Study [74], the risk of AMI was 5.9 times higher within 1 h of periods of vigorous-to-high-intensity PA compared with periods of lower levels of activity or rest. The RR of AMI was highest among those who exercised < 1 time a week (RR, 107) compared with those who exercised ≥ 5 times per week (RR, 2.4) (Figure 4) [75]. Accordingly, just 1–2 vigorous exercise sessions per week reduced the risk for exertion-related AMI by nearly 80%.

To put these data in perspective, consider the individual who participates in vigorous 1-h sessions of exercise ≥5 days per week. Using data from the Onset study [74], the risk of AMI during or soon after vigorous physical exertion is approximately doubled for this individual. Furthermore, regular exercise has been reported to reduce the long-term risk of cardiovascular events by up to 50% [76,77]. Thus, during the 1-h period of vigorous exercise, the individual’s risk will double and reach the level that it would have been at all times for his or her sedentary counterparts. However, during the remaining 23 h of the day, his or her risk would be ≤50% lower. Thus, there is a clear net benefit to regular exercise despite the modest transient increased risk of exercise-induced AMI [78].

## 4. Common Activities Associated with Acute Cardiac Events

Physical activities involving sudden bursts or high levels of anaerobic metabolism may transiently increase the risk for exercise-related acute cardiac events. These include downhill skiing [79], racquet sports [80], high-intensity interval training [81], and competitive sports activities (e.g., basketball) [82] as compared with other more moderate activities. Neural and psychologic stimuli secondary to competition may simultaneously increase sympathetic activity and catecholamine levels and lower the threshold for threatening ventricular arrhythmias [83]. Other recreational and domestic activities associated with increased cardiac demands and a greater incidence of cardiovascular events include deer hunting [84] and snow removal [85,86], as well as a marathon running [87] and triathlon participation [88].

*Marathon running*. The RACER study (Race Associated Cardiac Event Registry) evaluated the incidence and outcomes of cardiac arrest associated with marathon and half-marathon races in the U.S. over a 10.5-year period. The study population included 10.9 million registered runners (mean ± SD age, 42 ± 13 years) [87]. Of the 59 cases of cardiac arrest, 42 (71%) were fatal. The incidence rate was 3.75 times higher during full-marathons versus half-marathons and 5.6 times higher among men than women. Nearly half of all SCDs occurred during the final mile. The overall risk of a cardiac event during marathons and half-marathons was relatively low, versus other competitive endurance activities. Autopsy findings revealed that HCM and atherosclerotic CVD were the most common underlying anomalies.

*Triathlon participation*. The frequency of cardiac arrest and SCD has also been reported in > 9 million triathlon participants, 1985 to 2016 [88]. There were 135 SCDs, or 1.74 per 100,000 participants, which exceeded the incidence rate previously reported for marathon running (1.01 per 100,000 participants) [87]. The incidence of cardiovascular events was also 3.5-fold less in women than men. Most SCDs occurred during the swim segment (67%), whereas the remaining fatalities occurred during bicycling, running, and postrace recovery segments, respectively. Race experience was available for 68 participants, of whom 26 (38%) were competing in their first triathlon. Autopsies were performed on 61 victims, 27 (44%) of whom had either atherosclerotic coronary disease or cardiomyopathy.

In aggregate, these data suggest that cardiac arrest and SCD during marathon running and triathlon participation occasionally occur and that clinicians evaluating race participants should be aware of the heightened risks of HCM and atherosclerotic coronary disease in this patient population [89], both of which can often be detected with appropriate medical screening. An increased risk among “first-time” triathlon participants suggests inadequate preparation or poor training as potential contributors to some of the exertion-related fatalities [88]. Finally, participants should also be advised to heed warning symptoms and to avoid sprinting during the concluding minutes of the race, when cardiovascular events are most prevalent [90]. Symptomatic athletes should be strongly counseled to stop training and competing until medical assessment and clearance are obtained.

## 5. High-Volume, High-Intensity Endurance Training: Too Much of a Good Thing?

Among individuals who participate in high-volume, high-intensity endurance training regimens, use of cardioprotective medications are lower than among their less physically active counterparts [91]. These data, coupled with reports documenting the impressive risk factor profiles and superb cardiac performance of marathon runners, as well as the anti-aging effects that regular endurance exercise provides [92], have led an increasing number of health enthusiasts to embrace the notion that “more exercise is invariably better [93].”

Long-term conventional endurance or isotonic exercise training alters cardiac structure and function, and such adaptations are believed to be benign. These include: enlargement of all cardiac chambers; improvement in ventricular compliance and distensibility; and, electrical remodeling, such as sinus bradycardia, sinus arrhythmia, and first-degree atrioventricular block. Emerging evidence, however, suggests that over time, high-volume, high-intensity, exercise training regimens can induce cardiac maladaptations such as an increased incidence of atrial fibrillation (AF) and accelerated coronary artery calcification (CAC) [89]. Accordingly, there is debate whether intensive exercise can be harmful to the heart, especially in some individuals.

### 5.1. Atrial Fibrillation (AF)

Numerous epidemiologic and observational studies have reported a statistically significant association between chronic high-volume, high-intensity aerobic exercise training and a heightened risk of developing AF [89]. One analysis found that the risk of AF was 2- to 10-fold higher in endurance athletes than in matched controls [94]. Moreover, long-term vigorous endurance exercise (i.e., ≥2000 h of endurance training or ≥20 years of training) was strongly associated with an increased risk for lone AF [95,96]. Potential factors triggering AF in previously healthy long-term endurance exercisers include ventricular hypertrophy, which may cause some degree of diastolic dysfunction and increased left atrial stretch, enlargement or fibrosis, systemic inflammation, and increased vagal tone, potentially shortening the atrial refractory period (Figure 5) [89]. Fortunately, much of the risk for AF seems to resolve with detraining and/or reducing the exercise volume or intensity, probably in part because of the normalization of autonomic tone [97].

### 5.2. Accelerated Coronary Artery Calcification (CAC)

Several investigators have reported that veteran endurance athletes have a higher prevalence of elevated CAC scores when compared with age, gender, and risk factor matched control groups from the general population [98,99,100]. However, these athletes also demonstrate greater levels of CRF, expressed as METs, coronary artery size, and dilating capacity, as well as predominantly stable calcified plaques and fewer vulnerable mixed plaques than their less-active counterparts [99,100]. These concomitant adaptations appear to offset the negative impact of a higher CAC [89]. A landmark 8.4-year follow-up study of 8425 men from the Cooper Center Longitudinal study revealed that after adjusting for CAC level, each additional 1 MET increment in CRF conferred an 11% lower risk for subsequent cardiovascular events [101]. Collectively, these data suggest that the risk for adverse cardiovascular events is lower in physically active people than their inactive counterparts with the same CAC score. These results also provide reassurance that highly active individuals with CAC can safely continue their exercise programs, provided that they remain asymptomatic.

## 6. Prevention of Triggered Cardiac Events

There is now considerable evidence that acute cardiovascular events can be triggered by varied physical, chemical, and psychological stressors, including strenuous physical exertion [78]. The underlying mechanisms may involve biomechanical, prothrombotic, and arrhythmogenic stimuli, largely mediated by the associated heightened output of the sympathetic nervous system.

Fortunately, regular exercise, stress management, quitting smoking, and favorably modifying other coronary risk factors may attenuate the response to and protect against trigger-induced coronary events [102]. The beneficial role of regular moderate-to-vigorous exercise may be due to multiple mechanisms, including cardiovascular, neurologic and biochemical adaptations, as well as psychologic effects (Figure 6). Concomitant favorable autonomic adaptations include increases in heart rate variability, a strong prognostic indicator that is inversely related to mortality [103]. In addition, exercise preconditioning provides immediate cardioprotective benefits against AMI, by conferring transient antiarrhythmic and anti-ischemic effects against ischemic injury [104,105]. The impact of even brief bouts of PA that achieve a minimum threshold of ≥50% of functional capacity appear to provide the stimulus for improved clinical outcomes following acute cardiac events [105].

### Prophylactic Use of Cardioprotective Medications before Strenuous Exercise

Although some have suggested that recreational athletes with known or suspected CVD may benefit from taking aspirin or β-blockers shortly before competitive exercise, there are no definitive data indicating that these agents (despite their proven effectiveness for secondary prevention) decrease exercise-related acute cardiac events [89,106]. Accordingly, related reports [74,87] and the INTERHEART study [107] suggest there is insufficient evidence to recommend the prophylactic use of these medications before strenuous PA or competitive sports participation.

In summary, although there are varied potential strategies to reduce the risk of triggered acute cardiac events, we believe that increased lifestyle PA, structured exercise, or both, should be included in stress reduction interventions [78], due to the associated autonomic adaptations and parasympathetic predominance [108]. This strategy is readily accessible, beneficial for both physical and mental health, and provides up to eight-year gains in life expectancy in the most physically active population cohorts [109].

## 7. Clinical Recommendations

Previously sedentary patients with and without CVD should be encouraged to participate in gradually progressive exercise regimens. Because the least active, least fit individuals are at greatest risk for exercise-related acute cardiac events, they should follow a gradual increase in exercise intensity and avoid vigorous exercise (≥60% functional capacity or ≥6 METs), at least over the initial six to eight weeks. To promote and maintain health, moderate-intensity aerobic (endurance) PA (40–59% functional capacity or 3.0–5.9 METs) for a minimum of 30 min on five days each week, or vigorous intensity aerobic PA for a minimum of 20 min on three days each week, is recommended [110]. Patients with CVD seeking to greatly increase their PA intensity or volume should be evaluated and advised in accordance with contemporary guidelines [111].

## 8. Conclusions

The effects of chronic stress and varied stressors (i.e., physical, chemical, psychologic) in the development of CVD and triggering of acute cardiac events are well documented. Regular moderate-to-vigorous PA, structured exercise, and higher levels of CRF appear to be therapeutic in combatting the deleterious impact of chronic stress and other risk factors that are the forerunners of CVD. On the other hand, studies suggest that unaccustomed vigorous-to-high intensity physical exertion can trigger acute cardiovascular events, especially in habitually sedentary individuals with known or occult CVD. It is also reported that in some individuals, high-volume, high-intensity exercise regimens can, over time, lead to maladaptations, including increased coronary calcification and the development of AF, as described by a reverse J-shaped curve. Although exercise is widely acknowledged for its salutary effects, increasing data suggest that “it may be possible to get too much of a good thing”.

## Figures and Tables

**Figure 1 ijerph-18-09922-f001:**
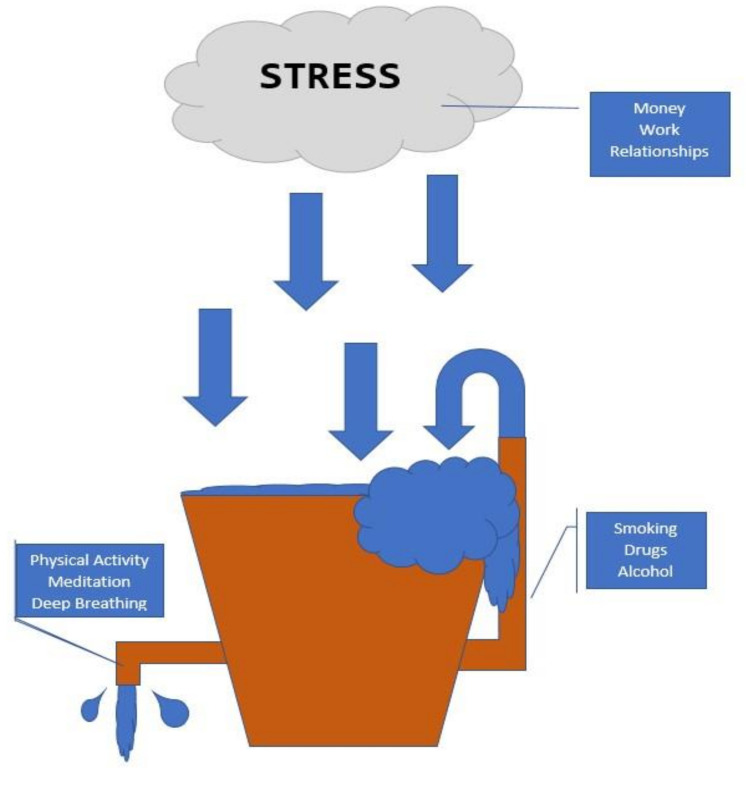
Stress bucket analogy. Chronic stress flows into the bucket.

**Figure 2 ijerph-18-09922-f002:**
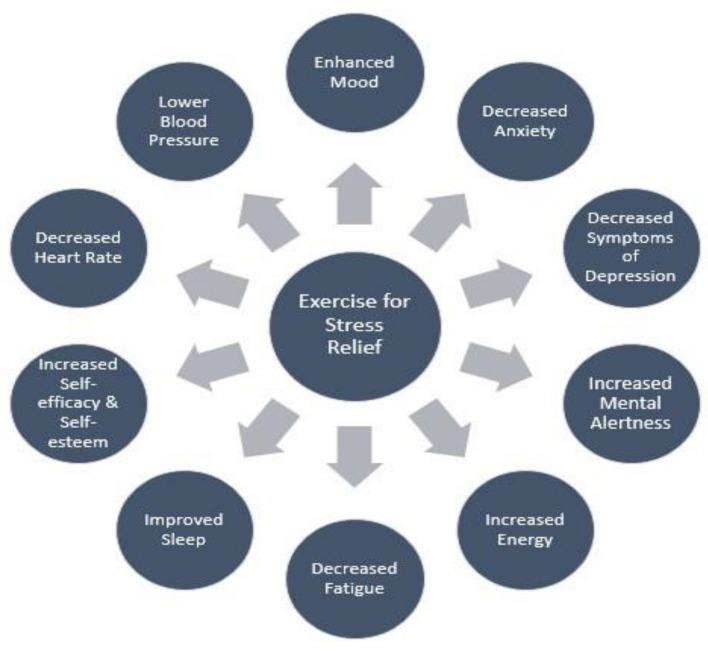
Potential stress reducing effects of regular lifestyle physical activity, structured endurance exercise, or both.

**Figure 3 ijerph-18-09922-f003:**
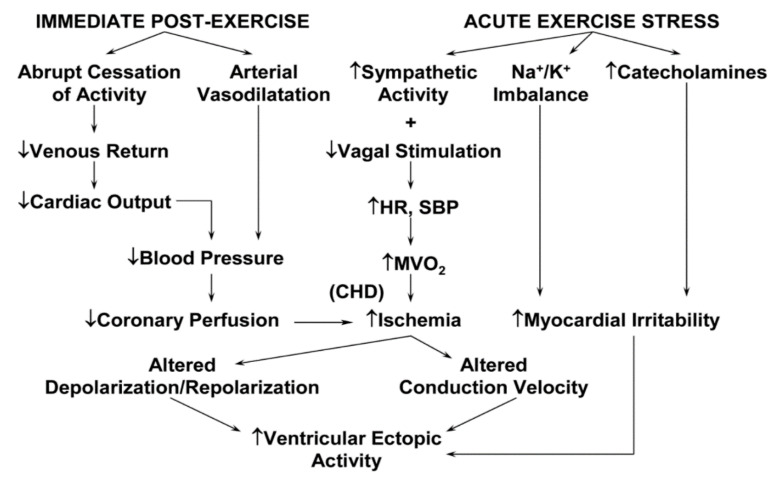
Physiologic alterations accompanying acute exercise training and recovery and their possible sequelae. CHD, coronary heart disease; HR, heart rate; SBP, systolic blood pressure; MVO_2_, myocardial oxygen consumption.

**Figure 4 ijerph-18-09922-f004:**
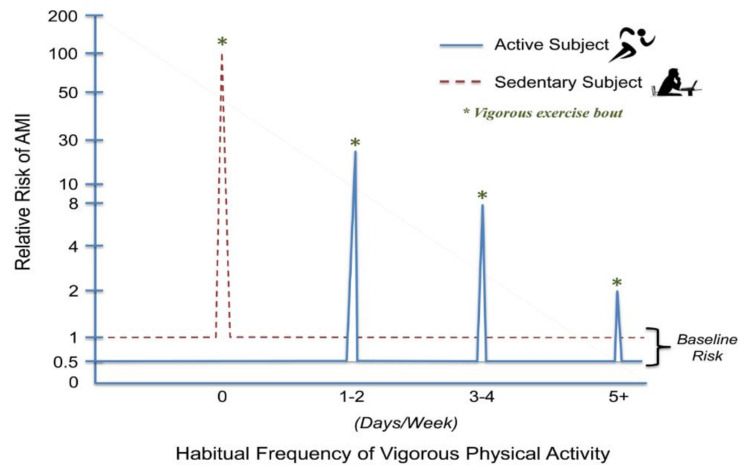
The relative risk for an acute myocardial infarction (AMI) is presented at rest (baseline) and during vigorous activity (days/week). No regular vigorous physical activity elevates the risk of an isolated, unaccustomed bout of vigorous exercise by ≥100-fold. Performing one to two days of weekly vigorous physical activity markedly lowers AMI risk during vigorous physical exertion. More frequent performance of vigorous physical activity further lowers exercise related AMI risk, although the relationship is not linear [74,75]. Vigorous exercise is delineated by the asterisks (*). Reprinted with permission from the American Heart Association.

**Figure 5 ijerph-18-09922-f005:**
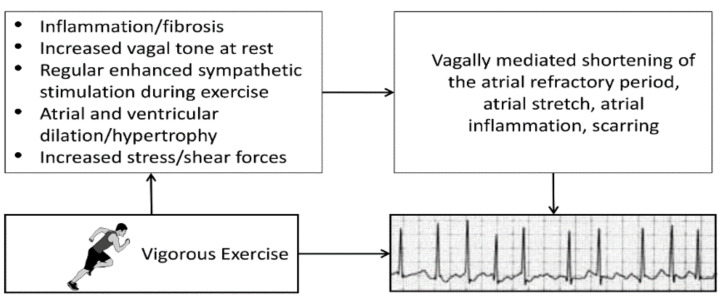
Potential mechanisms and associated sequelae for atrial fibrillation induced by strenuous endurance exercise. Provided with permission from the American Heart Association.

**Figure 6 ijerph-18-09922-f006:**
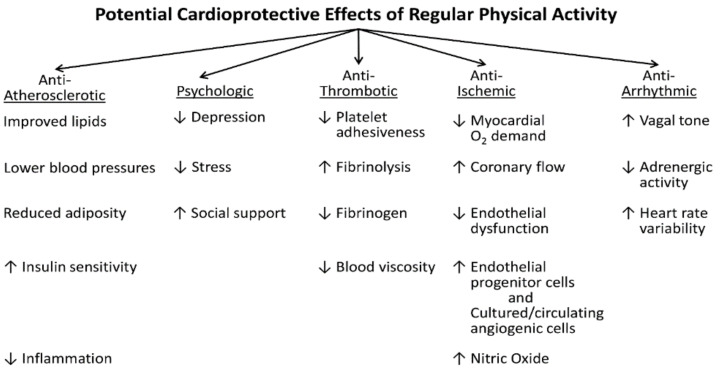
Multiple mechanisms by which moderate-to-vigorous physical activity may reduce the risk of initial and recurrent cardiovascular events. ↑, increased; ↓, decreased; O_2_, oxygen.

**Table 1 ijerph-18-09922-t001:** Potential triggering mechanisms of acute myocardial infarction by strenuous physical exertion.

Induces plaque rupture via:
• Increased heart rate, blood pressure, and shear forces
• Altered coronary artery dimensions
• Exercise-induced spasm in partially obstructed artery segments
Renders a fissured plaque more thrombogenic by:
• Deepening the fissure
• Increasing thrombogenicity
Induces thrombogenesis directly via:
• Catecholamine-induced platelet aggregation

## Data Availability

Not applicable.

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
