# Peer review of "Chronic Stress, Exercise and Cardiovascular Disease: Placing the Benefits and Risks of Physical Activity into Perspective"

_ijerph, 2021, doi:10.3390/ijerph18189922_

Round 1

Reviewer 1 Report

This review article addressed a very relevant topic: the integration of different physiological systems in determining the health benefits of physical activity and, by converse, the risks of inappropriate physical activity on health.

Major life stress, socioeconomic factors and chronic psychiatric conditions and their interacting mechanisms and their effects on cardiovascular disease and mortality are discussed quoting the most relevant studies from literature.

The role of physical activity in these complex interactions as well as the acknowledged and potential mechanisms are well delineated.

All the above arguments are well balanced and discussed judiciously

I have no further comments for the authors

Author Response

Response: Thank you for the very positive response to our comprehensive review of this timely and clinically relevant topic.

Reviewer 2 Report

nice review paper on a topic deserving more attention by clinicians. The manuscript is huge and treats several aspects even if different approaches and precision are adopted according to which part authors discuss.

Authors altern deep description, including molecular pathway, and narrative without a consistent style even if the overall readability is sufficient.

I suggest to try making more homogeneous the manuscript.

In addition, the manuscript touches more on the stress aspect and the impact of physical activity on it and less on the impact of exercise on cardiovascular disease. I suggest better defining the type of exercise (isotonic, isometric and resistance) and their impact on oxygen uptake, cardiac parameters (ie cardiac output) and vascular resistances. The definitions used in the manuscript (i.e. high volume, high intensity, vigorous) should be better specified in their meaning and reference values.

An item, potentially a table, to resume items and instruments to measure the stress (chronic and acute) could be very helpful. 

I suggest improving the distinction, in each paragraph, between the populations authors refer to, if with or without known cardiovascular disease, given the crucial differences in physical activities threshold (please see ESC guidelines 2020).

Few considerations were reported regarding the peak of oxygen consumption (VO2) across the manuscript. Please improve this aspect in light of the essential information provided by this parameter, if available, in the setting described by authors.

Minor issues:

  • table 1 could be modified in a bullets list, in order to improve the readability.
  • Figure 1: legend and paragraph 2.2 is pretty the same, please select which is to be maintained.
  • Figure 4: please enlighten better the vigorous exercise bout.

Author Response

I suggest better defining the type of exercise (isotonic, isometric and resistance) and their impact on oxygen uptake, cardiac parameters (i.e. high volume, high intensity, vigorous) should be better specified in their meaning and reference values.

Response: When possible, we have further defined the type of exercise we are referring to, particularly in sections 3 ̶ 8 of the manuscript.

An item, potentially a table, to resume items and instruments to measure the stress (chronic and acute) could be very helpful.

Response: We do not feel that this is necessary, as the items and instruments to measure chronic stress are already delineated (referenced) in sections #1 and 2 of the manuscript. Moreover, other reviewers have implied that the manuscript may already be too long (“the manuscript is huge”) and it now contains 8 sections, 1 table, 6 different figures, and 111 reference citations!

Few considerations were reported regarding the peak of oxygen consumption (VO2) across the manuscript. Please improve this aspect in light of the essential information provided by this parameter, if available, in the setting described by authors.

Response: To address the reviewer’s previous point regarding definitions and the importance of peak oxygen consumption, which in our review is synonymous with cardiorespiratory fitness (CRF), we have expanded the verbiage under section #2, titled, “Exercise/Physical Activity: A Double-Edged Sword.”

Minor issues

  • Table 1 could be modified in a bullets list, in order to improve the readability.
  • Figure 1: legend and paragraph 2.2 is pretty the same, please select which is to be maintained.
  • Figure 4: please enlighten better the vigorous exercise bout.

Response: Per the reviewer’s request, all 3 of these “minor issues” have now been rectified via changes in the revised manuscript text, figure legends, and Table. Thank you for these perceptive suggestions!

Reviewer 3 Report

This a very nice review paper on stress, exercise and CV disease. It is well structured and clearly covers the topic.

I would recommend to include a section on "Clinical recommendations", regarding exercise and stress in CV patients or in general population to prevent CVD, besides conclusions. 

Author Response

Response: To address the reviewer’s suggestion, we’ve now provided a brief additional section to this already lengthy manuscript. The section is now #7 and is titled “Clinical Recommendations,” and includes two (2)

Reviewer 4 Report

This is a well written, well researched manuscript on a timely and important topic. I found the writing style to be reader friendly and the figures helpful in further explaining the concepts. I couldn't find any grammatical, spelling or sentence structure errors. Perhaps for a bit more well roundedness, more could be noted about the role of heart rate variability as a stress marker. It is a broad topic though, so understandable that not everything can be covered.

Author Response

This is a well written, well researched manuscript on a timely and important topic. I found the writing style to be reader friendly and the figures helpful in further explaining the concepts. I couldn’t find any grammatical, spelling or sentence structure errors. Perhaps for a bit more well roundedness, more could be noted about the role of heart rate variability as a stress marker. It is a broad topic though, so understandable that not everything can be covered.

Response: We appreciate the reviewer’s generous comments regarding our comprehensive review and, to address his/her point on the important role of heart rate variability as a stress marker and prognostic indicator, we’ve added a statement to this effect in section #6 (Prevention of Triggered Cardiac Events). We’ve also cited Dr. Iellamo’s seminal reference on this topic (#103), as the citation here.

Round 2

Reviewer 2 Report

Authors have address all comments previously raised. No further considerations.
PS: Please note the the suggestion of a table for acute and chronic stress aimed to replace part of the (long) text section.